# Numerical and Experimental Study on Carbon Segregation in Shaped Billet of Medium Carbon Steel with Combined Electromagnetic Stirring

**DOI:** 10.3390/ma16237464

**Published:** 2023-11-30

**Authors:** Pengchao Li, Guifang Zhang, Peng Yan, Nan Tian, Zhenhua Feng

**Affiliations:** 1Faculty of Metallurgical and Energy Engineering, Kunming University of Science and Technology, Kunming 650093, China; 13696392999@163.com (P.L.); tian1852558@163.com (N.T.); fengzhenhua666@126.com (Z.F.); 2Linyi Iron and Steel Investment Group Special Steel Co., Ltd., Linyi 276000, China; 3Key Laboratory of Clean Metallurgy for Complex Iron Resources in Colleges and Universities of Yunnan Province, Kunming University of Science and Technology, Kunming 650093, China

**Keywords:** continuous casting, shaped billet, combined electromagnetic stirring, multi-physical field model, carbon segregation

## Abstract

Carbon segregation is the major and classical internal defect in the continuous casting process of carbon steel. Based on the combined electromagnetic stirring equipment for new billet in a steel plant, China, the influence of combined electromagnetic stirring (M-EMS + F-EMS) on the carbon segregation of 300 mm × 340 mm special-shaped billet was studied via numerical simulation and on-site industrialization tests. The Lorentz force and carbon solute distribution were simulated under different EMS parameters. The formation mechanism of the carbon segregation of medium carbon steel with different combined electromagnetic stirring processes was analyzed. The results show that: (1) with the combined action of “solute flushing” effect and gravity, the carbon concentration in the loose side of the medium carbon steel casting billet is gradually lower than the fixed side, while the carbon concentration on the fixed side gradually accumulates more; and (2) under the action of combined electromagnetic stirring, the segregation index of casting billet could be controlled to remain between 0.96–1.05 and shows an increasing change in solidification from the skin to the center. When the current and frequency of M-EMS are 250 A and 2.0 Hz and the F-EMS are 180 A and 8.0 Hz, the carbon segregation defects in the special-shaped (300 mm × 340 mm) casting billet can be significantly improved.

## 1. Introduction

Carbon segregation is one of the key indicators of continuous casting billet quality standards [1,2]. This defect will adversely affect the drawing stability during continuous casting, harming the uniformity of the properties of rolled and finished products [3,4]. Then, the control of carbon segregation is one of the key constraints on the development and application of high-quality steel [5]. In the rolling and subsequent heat treatment process, carbon segregation defects can only be reduced but cannot be eliminated [6,7]. Therefore, it is particularly important to improve the macro-segregation of continuous casting billet during the continuous casting stage. At present, most of the research on carbon segregation in continuous casting billet is focused on central segregation [8,9,10]. For continuous casting billet, a large part is used to replace steel ingots to produce large forgings, which have higher requirements for the performance uniformity of the entire section. Therefore, the control of the carbon segregation of continuous casting billet should focus on the fluctuation of carbon elements in the entire section.

With the maturity of electromagnetic stirring (EMS) technology, EMS technology has been widely used in the control of solidification structure and segregation behavior of continuous casting billet [11]. Luo et al. [12] conducted an industrial test on the function of final electromagnetic stirring (F-EMS) on the center segregation of the billet, and the results showed that the excellent billet quality was achieved when the current of F-EMS was 360–380 A. Sun et al. [13] adopted alternating F-EMS to make the internal organization of continuous casting billets more uniform and the carbon segregation index closer to 1 and more stable. Jiang et al. [14] studied the direction of EMS. When the EMS always moves in one direction, the improvement in the central segregation is better than that of the positive and negative alternating agitation. Du et al. [15] designed three combinations of EMS to investigate its influence on the quality of casting billet, using mold electromagnetic stirring (M-EMS) alone, M-EMS + F-EMS, and M-EMS + alternating F-EMS. Li et al. [16] used the same method to compare the effects of M-EMS alone and M-EMS + F-EMS on the internal quality of continuous casting billets and obtained the result that the combined EMS is much better than that of M-EMS alone. Ayata et al. [17] compared the low cross-section structure and the segregation of continuous casting billets without EMS, using M-EMS alone, strand electromagnetic stirring (S-EMS) alone, and M-EMS + F-EMS combined. For the application of EMS to the liquid steel in the continuous casting process, it is necessary to study the parameters in depth to improve the carbon segregation, which has a positive role in improving the mechanical properties and service performance. Dong et al. [18] obtained a typical macro-segregation mode, including the peak value of positive segregation on the center line and the minimum value of negative segregation on both sides through the proposed three-dimensional macro segregation model. The simulation results are in good agreement with the experimental results. Sun et al. [19] investigated the negative segregation behavior of the subsurface layer, and the effect on the compositional homogeneity of cast billets with a cross-section size of 320 mm × 425 mm was simulated.

In this paper, based on the continuous casting production of special-shaped medium carbon steel (300 mm × 340 mm), a coupled numerical model, encompassing electromagnetism, heat, and solute transport, was developed to investigate carbon segregation behaviors under different stirring parameters of combined electromagnetic stirring (M-EMS + F-EMS). Subsequently, combined with on-site industrialization tests, the operating parameters were optimized for the production of medium-carbon steel continuous casting in a steel plant, China.

## 2. Model Descriptions

### 2.1. Assumptions

In the present research, a number of hypotheses are proposed to simplify the complexity and enhance computational efficiency.

(1)The molten steel in the billet is regarded as an incompressible Newtonian fluid.(2)The transport phenomena occurring during the continuous casting process, i.e., the casting speed, the casting temperature, etc., are treated as steady states and remain unchanged [20].(3)Vibration, mold taper, bulging [21], and solidification shrinkage deformation influence on the flow field is ignored, and the meniscus was assumed for the adiabatic plane.(4)The solidification process in the liquid–solid interface is considered to occur in the local thermodynamic equilibrium state.(5)The thermal effect of F-EMS on the casting billet was ignored.

### 2.2. Governing Equations

Continuous casting electromagnetic stirring is a process in which a magnetic field is generated inside liquid steel. Then, the electromagnetic force is generated by the movement of the liquid steel to promote liquid steel movement [22]. Proper forced flow can effectively control the vertical growth of columnar crystal structure and does not cause obvious segregation [23]. Micro-segregation refers to the uneven distribution of solute elements between dendrites, which is caused by the different solubility of elements in the solid–liquid phase in the solidification process.

(1)Electromagnetic Model

The electromagnetic force for electromagnetic stirring can be obtained by calculating Maxwell’s equations, namely (1), (2), (3), and (4):(1)∇×H=J
(2)$$\nabla \times E=-\frac{\partial B}{\partial t}$$
(3)∇·B=0
(4)J=σ·E

In Equations (1)–(4), ***H*** represents the magnetic field strength, A/m; ***J*** represents the current intensity, A/m^2^; t denotes time, s; ***E*** denotes the electric field intensity, V/m; ***B*** denotes the magnetic flux density, T; and σ denotes the electric conductivity, S/m.

Then, the electromagnetic force expression [24] is shown as follows:(5)$$F_{m}=\frac{1}{2}Re(J\times B^{*})$$
where ***F_m_*** denotes the time-average electromagnetic volume force, *Re* denotes the real part of the complex quantity, and $B^{*}$ is the complex conjugate of ***B***.

(2)Fluid Flow Model

The process is considered to be steady; therefore, the partial time derivatives of the velocity are zero and can be deleted from the equations of continuity and momentum conservation. Then, the melt flow pattern is described via the continuity and momentum equations as follows:

Equation of continuity:(6)$$\nabla \cdot \left(\rho u\right)=0$$

In Equation (6), ρ is the fluid density (kg/m^3^) and u is the fluid velocity, m/s.

Momentum conservation [25] equation:(7)$$\nabla \cdot \rho \left(uu\right)=\nabla \cdot \left(\mu_{eff}\nabla \cdot u\right)-\nabla P+\rho g+S_{p}+F_{m}{+F}_{B}$$

In Equation (7), P is the static pressure, N/m^2^; $\mu_{eff}$ is the effective viscosity coefficient, Pa·s; g denotes the acceleration of gravity, m/s^2^; $F_{B}$ denotes the thermal buoyancy source term; and $S_{p}$ represents the Darcy source term, which can be calculated using the following equation.
(8)$$S_{p}=\frac{{\left(1-f_{l}\right)}^{2}}{f_{l}^{3}+0.01}A_{mush}\left(u-u_{s}\right)$$
where $u_{s}$ is the casting speed, m/s, and $f_{l}$ is the liquid fraction. In order to better describe the solute transport process, the low Reynolds number equation k−ε is used in this paper.
(9)$$\nabla \cdot \left(\rho uk\right)==\nabla \cdot \left[\left(\mu +\frac{\mu_{T}}{\sigma_{k}}\right)\nabla k\right]+G-\rho \varepsilon +S_{k}$$
(10)$$\nabla \cdot \left(\rho u\varepsilon \right)=\nabla \cdot \left[\left(\mu +\frac{\mu_{T}}{\sigma_{\varepsilon }}\right)\nabla \varepsilon \right]+C_{\varepsilon 1}G\frac{\varepsilon }{k}-C_{\varepsilon 2}\rho \frac{\varepsilon^{2}}{k}+S_{\varepsilon }$$
(11)$$\mu_{T}=\rho C_{\mu }\frac{k^{2}}{\varepsilon }$$

$S_{k}$ and $S_{\varepsilon }$ are source terms, which are adopted to take the effects of the porous material into consideration. The values of constants in the model are: $C_{\mu }=0.09$, $C_{\varepsilon 1}=1.44$, $C_{\varepsilon 2}$ = 1.92, $\sigma_{k}=1.0$, and $\sigma_{\varepsilon }=1.3$ [26].

(1)Heat Transfer Model

Heat transfer energy conservation equation:(12)$$\frac{\partial }{\partial t}(\rho H)+\nabla \cdot \left(\rho uH\right)=\nabla \cdot \left(K_{eff}\nabla T\right)$$

In Equation (12), H denotes the total enthalpy, J/kg, and $K_{eff}$ denotes the effective thermal conductivity, W/(m·K). H and $K_{eff}$ can be obtained by solving the following equation.
(13)$$H=h_{ref}+\int_{T_{ref}}^{T}c_{p}dT+f_{l}L$$
(14)$$K_{eff}=\left\{\begin{array}{c} K_{T,s}\;\;\;\;\;\;\;\;\;\;\;\;\;\;\;\;\;\;\;\;\;\;\;\;\;\;\;\;\;\;\;\;\;\;\;\;\;\;\;\;\;\;\;\;T\leq T_{s}\\ K_{T,s}f_{s}+K_{T,l}f_{l}\;\;\;\;\;\;\;\;\;\;\;\;\;\;\;\;\;\;\;\;\;\;\;\;\;\;\;\;T_{s}<T<T_{l}\\ K_{T,l}+\frac{\mu_{T}}{{Pr}_{t}}\;\;\;\;\;\;\;\;\;\;\;\;\;\;\;\;\;\;\;\;\;\;\;\;\;\;\;\;\;\;\;\;\;\;T>T_{l} \end{array}\right.$$

In Equation (13), $h_{ref}$ is the reference enthalpy, J/kg; $c_{p}$ denotes the specific heat capacity of steel, J/(kg·K); and *L* is the latent heat of steel, J/kg. In Equation (14), ${Pr}_{t}$ denotes the turbulent Prandtl number and $f_{l}$ denotes the liquid phase fraction, which can be obtained by solving the following formula:(15)$$f_{l}=\left\{\begin{array}{c} 0\;\;\;\;\;\;\;\;\;\;\;\;\;\;\;\;\;\;\;\;\;\;\;\;\;\;\;\;\;\;\;\;\;\;\;T\leq T_{s}\\ \frac{T-T_{s}}{T_{l}-T_{s}}\;\;\;\;\;\;\;\;\;\;\;\;\;\;\;\;\;\;\;\;\;\;\;\;\;\;\;T_{s}<T<T_{l}\\ 1\;\;\;\;\;\;\;\;\;\;\;\;\;\;\;\;\;\;\;\;\;\;\;\;\;\;\;\;\;\;\;\;\;\;\;T>T_{l} \end{array}\right.$$

(2)Solute Transfer Model

Solute conservation equation:(16)$$\nabla \cdot \left(\rho uc\right)=\nabla \cdot \left[\left(\rho D_{l}+\frac{\mu_{turb}}{S_{Ct}}\right)\nabla c\right]+S_{s}$$

In Equation (16), *c* is the concentration of solute element, wt.%, and $D_{l}$ is diffusion coefficient of solute in the liquid phase. $S_{Ct}$ is turbulent Schmidt number, and a value of 1.0. $S_{s}$ is usually the source term in the solute conservation equation, which includes the molecular diffusion term $S_{dif}$ and the convection–diffusion $S_{con}$ calculated via Equation (17) with Equation (18), respectively.
(17)$$S_{dif}=\nabla \cdot \left[\rho f_{s}D_{s}\nabla \cdot \left(c_{s}-c\right)\right]+\nabla \cdot \left[\rho f_{l}D_{l}\nabla \cdot \left(c_{l}-c\right)\right]$$
(18)$$S_{con}=\nabla \cdot \left[\rho \left(u-u_{s}\right)\left(c_{l}-c\right)\right]$$

### 2.3. Model Parameter and Numerical Solution

The schematic diagram of the EMS used in this paper is shown in Figure 1. The numerical model was obtained using the finite element software COMSOL Multiphysics v6.0. The model is composed of an electromagnetic agitator, a toroidal core, and 12 coil packages, the whole process model of continuous casting is divided into three sections: the first section ends at the secondary cooling zone; the second section is the second, third, and fourth section of the second cooling zone and the air cooling section before F-EMS (about Z = −9 m); the third section is the F-EMS mixing section until the casting billet is completely solidified, and the F-EMS installation position is Z = −9.34 m. The whole geometry model of the electromagnetic field was taken to be surrounded by an air sphere in which most of the magnetic flux lines are closed. Both the exit and the entrance of the model are velocity boundary conditions. The velocity at the exit is set to the drawing speed, and the velocity at the entrance is set to $u_{in}$; it can be calculated according to Equation (19).
(19)$$u_{in}=\frac{4L^{2}}{\pi d^{2}}u_{s}$$

In Equation (19), $L^{2}$ is the cross-sectional area of the casting billet, m^2^; d is the nozzle diameter, m. Set the meniscus to adiabatic. The wall surface of the outer surface of the billet is set as the slip boundary condition in the flow field calculation. When calculating the heat transfer in the part of the mold, the surface heat flux is set to q, and its magnitude can be calculated via Equations (20)–(22).
(20)$$q=2,680,000-b\sqrt{L/u_{s}}$$
(21)$$b=\frac{1.5\times \left(2,680,000-\bar{q}\right)}{\sqrt{L_{m}/u_{s}}}$$
(22)$$\bar{q}=\frac{c_{w}Q_{m}∆T}{S_{eff}}$$

In Equations (20)–(22), L is the distance to the meniscus, m; $L_{m}$ is the distance from the meniscus to the exit of the mold, m; $c_{w}$ is the specific heat capacity of cooling water, J/(kg·K); $Q_{m}$ is cooling water flow, m^3^/s; ∆T is the temperature difference between the inlet water temperature and the return water temperature of the cooling water, K; and $S_{eff}$ is the contact area between the billet and the mold, m^2^. The surface of the billet in the secondary cooling zone is set as convective heat transfer, and the heat transfer coefficient is determined by Equation (23). The surface of the billet in the air cooling zone is set as surface radiation to the environment, which is determined via Equation (24).
(23)$$h_{s}=116+10.44w^{0.851}$$
(24)$$q_{s}=\varepsilon \sigma \left(T_{b}^{4}-T_{a}^{4}\right)$$

The macro-segregation model was solved in the computational domain using a finite segmentation calculation method. Table 1 lists the relevant parameters of the continuous casting and molten steel. Considering the multiple streams on site, the casting speed is set to 0.65 m/min.

## 3. Experimental Implementation

The sampling diagram for carbon segregation analysis is shown in Figure 2. A 5 mm heigh sample with diameter 5 mm was removed from the billet section for analysis, and a total of nine samples were taken from the bloom (the length of the bloom is Y = 300 mm, and the width of the blank X = 340 mm. Five points are the central points, two points are the Y1/8 points, three points are the Y1/4 points, four points are the Y3/8 points, six points are the X1/8 points, seven points are the X1/4 points, eight points are the X3/8 points, and one and nine points are the 10 mm subcutaneous edges). Carbon concentration is analyzed using a carbon–sulfur analyzer (EMIA Pro, Horiba Inc., Kyoto, Japan). Medium carbon steel was used in this research, and the chemical analysis results are presented in Table 2.

In order to clearly express the variation of carbon concentration, the segregation index was used to quantify the macroscopic segregation inside the continuous casting billet. The carbon segregation index is presented via Equation (26) [29]:(25)$$a=\frac{C_{i}}{C_{l}}$$
where a denotes the carbon segregation index; $C_{i}$ denotes the carbon concentration at the sampling point *i*; and $C_{l}$ denotes the carbon concentration of molten steel in the tundish.

## 4. Results and Discussion

### 4.1. Model Validation

To verify the reliability of the macro-segregation of the multi-physical model established by this model, the carbon solute distribution results of the billet under the condition of continuous casting at 1788 K and casting speed of 0.65 m/min without EMS are presented in Figure 3a. The carbon segregation results were compared with the billet under the same continuous casting conditions, as shown in Figure 3b. As can be seen in the results of casting billet and simulation shown in Figure 3, negative segregation and positive segregation appear in the same position despite some deviation of carbon concentration. In addition, the deviations at 0.2–0.4 m under the skin of the square billet may be related to the fluctuation of influencing factors in the production process. Then, the comparison results indicate that the prediction of carbon distribution using the multi-physical model has good reliability.

### 4.2. Lorentz Force of Casting Billet with Electromagnetic Stirring

Lorentz force is the main driving force of electromagnetic stirring, which drives the liquid steel to rotate. Electromagnetic stirring functions under different currents and frequencies and does not affect the direction and distribution trend of Lorentz force. Figure 4 presents the Lorentz force distribution in the billet with the action of F-EMS under 250 A and 8 Hz.

As shown in Figure 4a, the F-EMS is installed at Z = −9.5 m, and the Lorentz force distribution is symmetrically distributed along both sides of the casting billet. The Lorentz force at the center is the largest, and the surface is 9700 N/m^3^. It can be seen from Figure 4b that the reverse of Lorentz force is clockwise rotation, the maximum Lorentz force on this section is at the corner of the square, and the Lorentz force decreases as the distance from the final electromagnetic stirring center gets closer. This is closely related to the “skin effect” of electromagnetic fields. In order to investigate the effect of F-EMS with different current and frequency on the internal flow field of the casting billet, the X-axis Lorentz force distribution at the casting blank center is shown in Figure 5.

As shown in Figure 5, the Lorentz force increases linearly with the distance increasing from the bloom center to the edge and approaches zero at the center. The Lorentz force increases with the increase in current, which agrees with Ampere’s law. In the continuous casting bloom, the most obvious area of electromagnetic stirring effect is the liquid phase region of bloom. As can be seen in Figure 5a, the maximum Lorentz force in the liquid phase section is 3300 N/m^3^ at a current of 250 A. Then, the maximum Lorentz force in the liquid phase section shown in Figure 5b is 1480 N/m^3^ at the frequency of 8.0 Hz. Therefore, the stirring current is more effective in improving the internal flow of the bloom than the frequency. This could guide the adjustment of EMS parameters to obtain excellent bloom quality.

### 4.3. C Solute Distribution in Casting Billet

The results of carbon solute distribution in the mold are shown in Figure 6. In the mold area, the solute is enriched below the meniscus partly due to the influence of eddy currents and hot solute buoyancy. The concentration of carbon in the liquid phase in the center of the crystallizer has not changed, and the concentration of carbon on the surface is the same as the concentration of liquid steel. At the lower end of the model, as the position moves inward, the positive segregation gradually changes to negative segregation, which is caused by the “washing effect” of the liquid steel on the solidification front and the strong cooling condition of the mold.

The distribution of solute carbon elements at different positions from the meniscus is shown in Figure 7. With the solidification process, the liquid phase cavity gradually decreases, and carbon does not simply accumulate in the liquid phase cavity on the cross-section. Due to the convection caused by gravity and hot solute buoyancy in the natural convection zone, the solute distribution in the liquid phase cavity is uneven. The liquid steel in the liquid phase cavity forms two symmetrical circulations on the cross-section due to the action of buoyancy and the influence of gravity, which leads to more complex macroscopic segregation behavior. As shown in Figure 7, under the dual action of the “solute scour” effect of circulation and gravity, the concentration of solute elements is gradually less than the outer arc side, while the solute elements gradually accumulate and increase on the outer arc side.

When the appropriate parameters of M-EMS are identified as a current of 250 A and a frequency of 2.0 Hz, 8 Hz is set as the electromagnetic stirring frequency at the end of solidification. Then, the currents are considered 180 A, 200 A, and 250 A, and the distribution of carbon solute in the billet after completely solidifying is shown in Figure 8. As shown in Figure 8a–c, the distribution of carbon in the casting billet changes significantly under the action of electromagnetic force generated using F-EMS. The concentration area of central carbon shifts from the center to the left with the clockwise rotation of the flow field, and the area of the concentration area is larger than that without F-EMS.

The effects of different currents on carbon segregation are shown in Figure 8d, which the sampling position is shown in black line. Because the frequency has a slight influence on the distribution of the flow field and electromagnetic force, the carbon concentration distributions on the center line are compared under the currents of 0 A, 180 A, 200 A, and 250 A. As shown in Figure 8d, the carbon solute concentration increases significantly in the center without EMS, resulting in a serious center segregation problem. With the enhancement of the current in F-EMS, the carbon center segregation problem was significantly improved. This is mainly attributed to the stirring action of F-EMS on the liquid phase hole near the installation position and the distribution of carbon becoming more uniform. However, the improvement of carbon solute segregation with different currents is not obvious, which is because the tangential velocity generated in the liquid phase hole does not significantly augment with the current increase.

### 4.4. Experimental Study on the Structure and Macro-Segregation in the Casting Billet

The solidification structure of the casting billets with M-EMS and combined EMS are presented in Figure 9 and Figure 10, respectively. The overall quality of the generous billet is very poor without M-EMS, as shown in Figure 9a. Then, there are some obvious corner cracks in the subcutaneous tissue, and these small corner cracks could not be eliminated during subsequent bar rolling. They expand easily, resulting in elongated cracks in the rolling billet. As can be seen in Figure 9b, the thick columnar crystals grow vertically upward along the bottom surface of each side, and it is easy to bridge the columnar crystals in the central part due to the central loose side and central shrinkage hole. In Figure 9a–c, a light pattern is formed in the equiaxed crystal area in the central area. This is called the “white band”, which is generally caused by negative segregation. With the continuous increase in M-EMS current intensity, the white band in the casting billet also disappeared. In particular, as marked in Figure 9d, the overall carbon segregation fluctuates the least under the parameter configuration of the 250 A current and 2 Hz frequency, and the segregation index is maintained at 1.06–1.18.

In order to further optimize the combined EMS parameters and study the influence of F-EMS on carbon segregation, an optimization test on F-EMS was carried out with the parameters of mold electromagnetic stirring at a current of 250 A and a frequency of 2.0 Hz, and the result is shown in Figure 10.

In Figure 10a,f, positive segregation appears at the center of the billet, and other points appear alternatively as positive and negative segregation, of which negative segregation forms more easily. In Figure 10b,f, when the final electromagnetic stirring current is 180 A, the carbon segregation index is controlled to remain within the limits of 0.96 to 1.05, accompanied by a narrower fluctuation, and the trend increases from the subcutaneous to the solidifying center. When the current increases to 200 A, the carbon segregation index is between 1.07–1.20, which gradually increases from subcutaneous to the solidifying center, where the carbon segregation is more serious. When the F-EMS increases from 220 A to 250 A, the carbon segregation index is controlled at between 1.04 and 1.3. Meanwhile, the trend of carbon segregation increases from the subcutaneous area to the solidifying center, resulting in serious carbon segregation at the billet center. According to the comprehensive analysis of Figure 10, under the combined electromagnetic stirring optimization parameters, the mold electromagnetic stirring current is 250 A with a frequency of 2.0 Hz, and the final electromagnetic stirring current is 180 A with a frequency of 8.0 Hz. Under this parameter, the probability of negative segregation can be significantly reduced.

## 5. Conclusions

(1)The simulation results of combined electromagnetic stirring reveal that a symmetric circulation pattern emerges on the transverse section of the casting square billet due to the effect of thermal and solute buoyancy in the natural convection zone. Under the influence of electromagnetic stirring at the end of square billet solidification, the variation in tangential velocity generated via electromagnetic stirring in the liquid phase pockets of the casting is minimal at different current levels, resulting in relatively minor differences in the distribution of carbon solute within the cross-section of square billet.(2)The 300 mm × 340 mm special-shaped billet industrial test samples for carbon segregation show that the appropriate current and frequency parameters for M-EMS are 250 A and 2.0 Hz, while the suitable current and frequency parameters for F-EMS are 180 A and 8.0 Hz. Under these conditions, the carbon segregation index is controlled in order to remain within the limits of 0.96 to 1.05.

## Figures and Tables

**Figure 1 materials-16-07464-f001:**
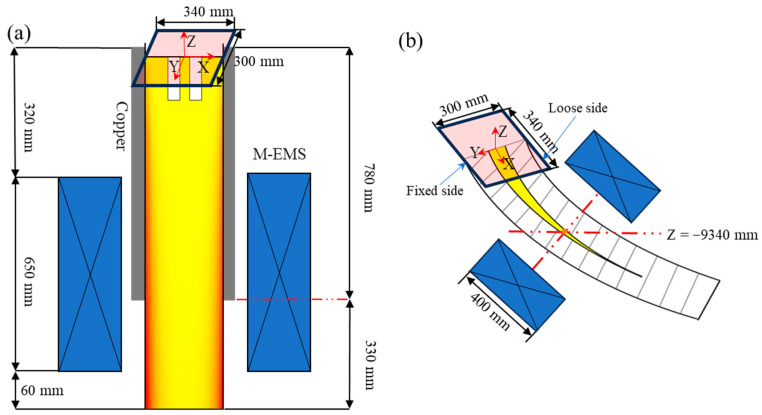
Schematic diagram of the combined EMS (mm): (**a**) M-EMS; (**b**) F-EMS.

**Figure 2 materials-16-07464-f002:**
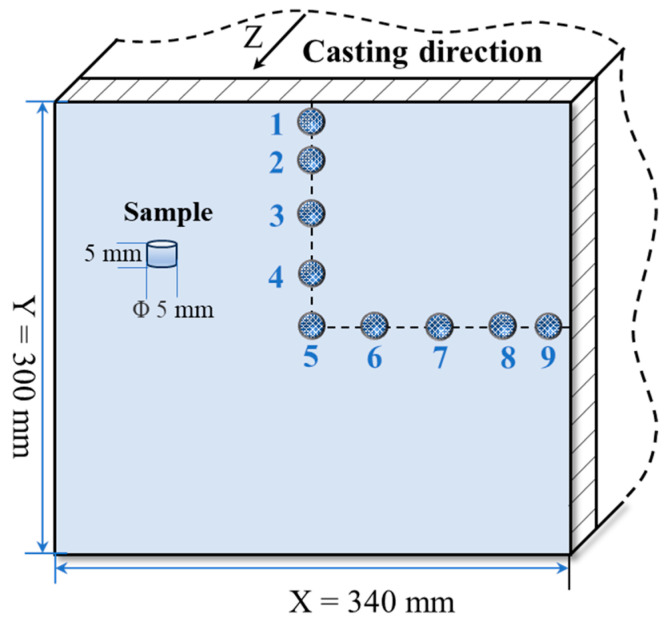
Diagram of sampling point for carbon content analysis.

**Figure 3 materials-16-07464-f003:**
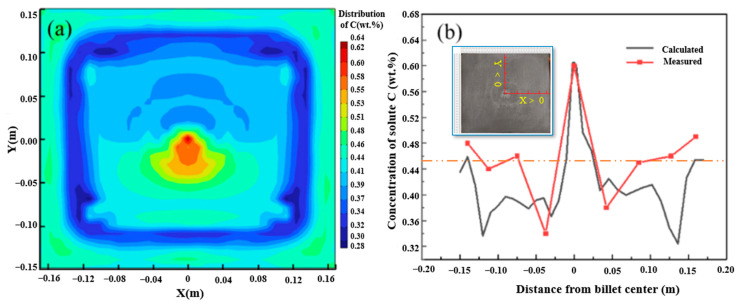
Verification of carbon segregation in casting billet: (**a**) simulation results; (**b**) verification results.

**Figure 4 materials-16-07464-f004:**
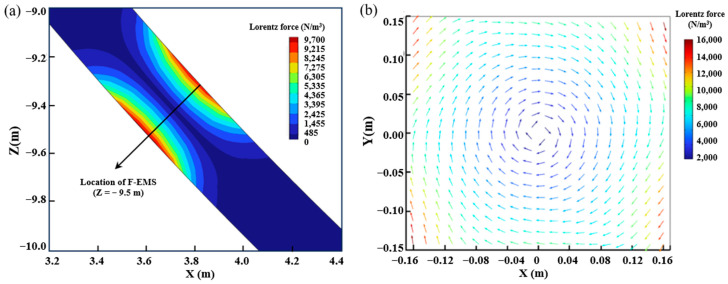
Lorentz force distribution in the area of F-EMS: (**a**) longitudinal section; (**b**) transverse cross-section.

**Figure 5 materials-16-07464-f005:**
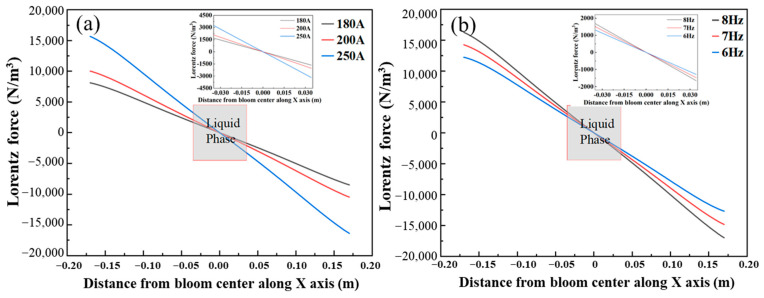
Lorentz force distribution in the area of F-EMS: (**a**) different current with frequency of 8.0 Hz; (**b**) different frequencies with current of 180 A.

**Figure 6 materials-16-07464-f006:**
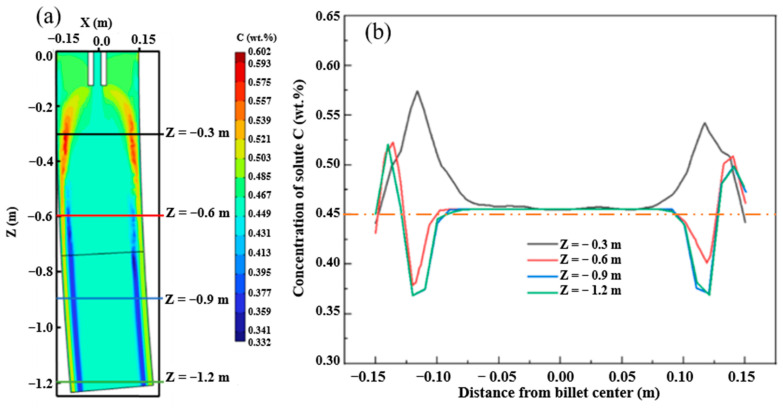
Carbon solute distribution in the central section of F-EMS: (**a**) carbon solute distribution in the central axial section; (**b**) distribution of carbon solutes at different locations.

**Figure 7 materials-16-07464-f007:**
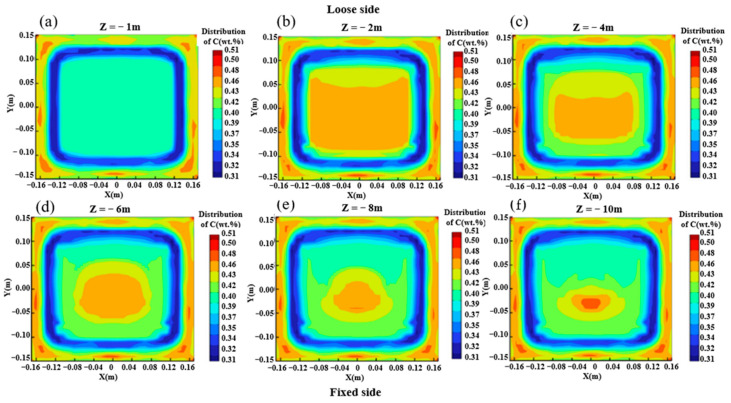
Distribution of carbon solutes at different distances from meniscus: (**a**) Z = −1 m; (**b**) Z = −2 m; (**c**) Z = −4 m; (**d**) Z = −6 m; (**e**) Z = −8 m; (**f**) Z = −10 m.

**Figure 8 materials-16-07464-f008:**
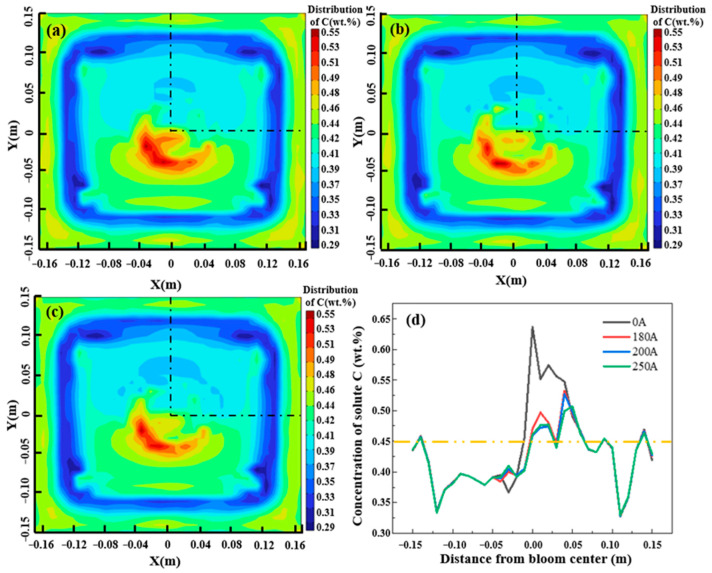
Distribution of carbon solutes with F-EMS at different currents: (**a**) 180 A; (**b**) 200 A; (**c**) 250 A; (**d**) line chart of carbon distribution.

**Figure 9 materials-16-07464-f009:**
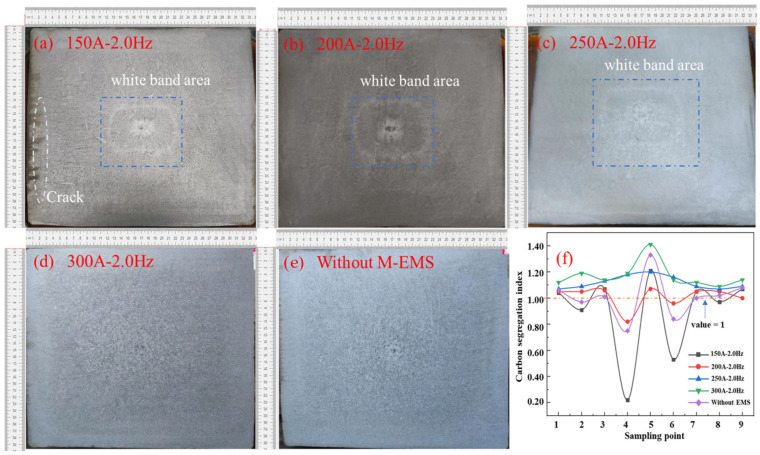
Effect of M-EMS current on the solidification structure of casting billet: (**a**) 0 A; (**b**) 150 A; (**c**) 200 A; (**d**) 250 A; (**e**) 300 A; (**f**) carbon segregation index.

**Figure 10 materials-16-07464-f010:**
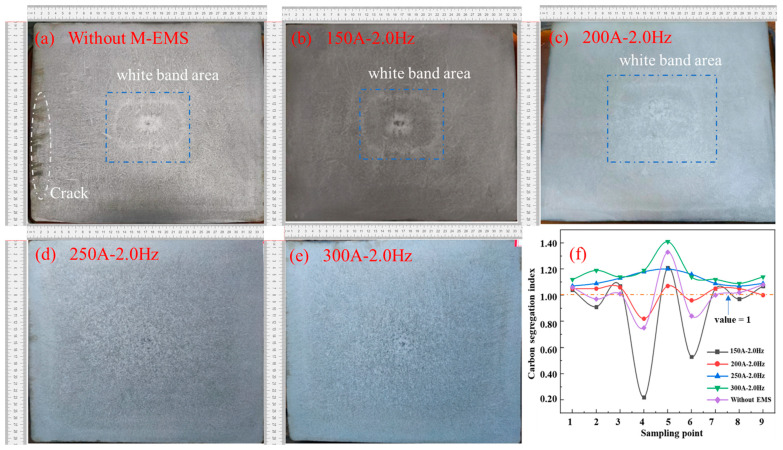
Effect of F-EMS current on the solidification structure of casting billet: (**a**) 160 A; (**b**) 180 A; (**c**) 200 A; (**d**) 220 A; (**e**) 250 A; (**f**) carbon segregation index.

**Table 1 materials-16-07464-t001:** Relevant parameter for continuous casting and molten steel [4,27,28].

Parameter	Value
Density/kg·m^−3^	7000
Casting speed/m·min^−1^	0.65
Casting temperature/K	1793
Liquid thermal conductivity/W·(m·K)^−1^	35
Solid phase thermal conductivity/W·(m·K)^−1^	38
Heat capacity at constant pressure/J·(kg·K)^−1^	800
Dynamic viscosity of liquid steel/Pa·s	0.006
Carbon solute concentration/wt.%	0.45
Latent heat of fusion/J·kg^−1^	250
Carbon diffusion coefficient in solid phase/cm^2^·s^−1^	0.0761exp($\frac{-134,557.44}{8.314\cdot T}$)
Carbon diffusion coefficient in liquid phase/cm^2^·s^−1^	0.0052exp($\frac{-11,700}{8.314\cdot T}$)
Current range of M-EMS/A	0, 150, 200, 250, 300
Frequency range of M-EMS/Hz	2.0
Current range of F-EMS/A	160, 180, 200, 220, 250
Frequency range of F-EMS/Hz	8.0

**Table 2 materials-16-07464-t002:** The main chemical components of medium carbon steel used in this study (wt.%).

Element	C	Cr	Mn	Ni	P	Si	Fe
Content	0.45	0.25	0.58	0.28	0.013	0.22	Bal.

## Data Availability

The data presented in this study are available on request from the corresponding author.

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
