# Peer review of "Numerical and Experimental Study on Carbon Segregation in Shaped Billet of Medium Carbon Steel with Combined Electromagnetic Stirring"

_materials, 2023, doi:10.3390/ma16237464_

Round 1

Reviewer 1 Report

Comments and Suggestions for Authors

Brief summary:

The topic is exciting, popular in the last decade, and important for the industry. The authors studied the influence of combined (Mold and Final) electromagnetic stirring on carbon segregation of 300mm × 340mm special-shaped medium carbon steel casting billet. Numerical simulation together with on-site tests in a Steel Plant in China were analyzed. The carbon concentration on the loose side appeared to become gradually lower, while the carbon concentration on the fixed side gradually accumulated more. Under the action of combined electromagnetic stirring, the segregation index could be controlled between 0.96-1.05, showing an increasing trend change from the skin to the solidification center. The recommended current and frequency values are 250A and 2.0Hz for Mold EMS, and 180A and 8.0Hz for Final EMS.

Broad comments:

There are many different papers dealing with the Electromagnetic Stirring of Steel aiming to improve carbon macro-segregation during casting. Those that present numerical investigation usually provide a set of partial differential equations describing the problem, although the authors mostly do not solve the equations themselves, they apply some commercial software to gain the solution. Still, this does not mean, that it is acceptable to write these equations with mistakes. The equations used in the submitted manuscript need thorough revision because there are too many errors in them. Although most of them are formal it is necessary to fix them before publication.

It is also necessary to inform the readers which program was used for numerical simulation.

Verification of the numerical model is not convincing, Figure 3 presents quite different values of concentration at distances -0.12 and 0.13; the calculated decrease of solute C concentration in these regions in the simulation should be explained.

English is fine, there are just a few items that have to be corrected; they are listed in the grammar part of the review and they are highlighted in blue in the main part of the revision.

All the references seem to be important for the topic, maybe some others should be mentioned as well, namely, https://doi.org/10.3390/met9090946 seems to be the origin of the correct version of the eq. (6).

The whole paper makes an impression that a special special-shaped sample of steel was examined and the results of this unique numerical modelling and measurement are summarized. However, it would be much more interesting for readers to generalize the findings and try to predict, if possible, which values of current and frequency should be useful for other types of sample shapes, different dimensions, or other types of steel. Is it possible to improve the conclusion part in such a manner?

Specific comments: Unfortunately too many, mostly formal, errors must be corrected.

62         The acronym S-EMS is not explained anywhere.

86-88    Steady or quasi-steady state should be defined better, the English in the sentence sounds strange, so changing the word order might be helpful.

102-111 Electromagnetic Model as a whole must be completely revised and corrected:

Both Maxwell equations (1)-(5) and the explanation of notation applied in them are not written properly, and basic rules of denoting physical characteristics were not complied.

105-107 H represents the magnetic field intensity, A/m; J represents the current intensity, A/m2; t denotes time, s; E denotes the voltage, V/m; B denotes the magnetic flux density, T; ... The characteristics should be written in italics in the text as well as in the equations, those that are vectors ( H, J, E, B) must be written in bold italics font, both in the equations and in the text. (H cannot be described as the magnetic induction intensity, because “magnetic induction” is just a synonym for “magnetic flux density”.)

109-110 Equation (6) must be completely rewritten: Fm, J, and B are vectors (written in bold italics), on the other hand, the real part of a complex number should be noted in straight letters, i.e. “Re”. The proper term is “real part” not “genuine part”. One of the first papers defining Lorentz force as the real part of the half cross product of current density and magnetic flux density mentioned that complex conjugate is used instead of magnetic induction B. Can you prove that your version is correct?

112-122 Fluid Flow Model should be commented on, the physical point of view is missing. (You can apply e.g. the following statement: “The process is considered to be steady, therefore the partial time derivatives of velocity are zero and they can be deleted from the equations of continuity and momentum conservation” as an introduction to the paragraph.)

There are serious errors on both sides of the eq.(8), vector ? must be outside the brackets, and the whole term should be ?(?⋅∇)? on the left-hand side of the equation, and the negative sign before the nabla operator must be deleted. Gravity acceleration and Darcy source are vectors (bold font necessary). Lorentz and buoyancy forces are missing on the right-hand side without explanation and still, they are discussed later in the text.

Eq. (12) is strange – what does the colon sign mean?

123-131 Heat Transfer Model also contains several errors – eq. (14) cannot be correct, the nabla operator on the left-hand side should be applied only on the enthalpy term. It cannot be correct to use the uppercase letter H, call it total heat enthalpy, and claim that its unit is J/kg. It should rather be simply specific total enthalpy (the word “heat” must be deleted).

126       ? and ?eff can be obtained by solving the following equation.” There is no equation to be solved, eq. (15) and (16) are just expressions for calculating the respective characteristics, therefore is should be better to write “? and ?eff can be obtained from the following equations”; the subscript must be added to K in the eq. (16), of course.

128       ref is the enthalpy of heat at reference heat” is completely nonsense, it should probably be the specific enthalpy at reference temperature. However, the reference temperature was neither specified, nor mentioned. This point should be clarified. Latent heat L in the text should be written in italics.

123-131 Solute Transfer Model must be revised as well. Concentration c must be written in italics (not bold!!!) both in the equations and in the following text.

135       Schmidt number should be written with capital letter in the beginning.

171       “A j5mm sample …” the text makes no sense. Based on Figure 2 it is clear that it should be modified in the following way: “A 5mm heigh sample with diameter 5mm …”

The carbon diffusion coefficient in solid phase/cm2·s-1 in Table 1 is probably wrong; according to reference [24] it should be 0.0761exp(−32160/8.314∙? ) instead of 0.0761exp(−134557.44/8.314∙? ).

206       "Across cross-section" should be replaced by "Transverse cross-section"

Figure 5               Neither the caption of the figure nor the surrounding text explains clearly what the four figures mean, the smaller ones in the upper part are unreadable, central region is the Liquid Phase (“u” is missing)

217       “Lorentz force increases linearly with the distance increases from the bloom center to the edge …“ should be corrected to “Lorentz force increases linearly with the distance increasing from the bloom center to the edge …“

Comments on the Quality of English Language

86-88    Steady or quasi-steady state should be defined better, the English in the sentence sounds strange, so changing the word order might be helpful.

206  "Across cross-section" should be replaced by "Transverse cross-section"

Figure 5 a,b   “u” is missing in the word Liqid Phase

217       “Lorentz force increases linearly with the distance increases from the bloom center to the edge …“ should be corrected to “Lorentz force increases linearly with the distance increasing from the bloom center to the edge …“ 

Reviewer 2 Report

Comments and Suggestions for Authors

This paper reports on the Numerical and Experimental Study on Carbon Segregation in Shaped Billet of Medium Carbon Steel with Combined Electromagnetic Stirring”. The subject is interesting, most parts of the manuscript appear to be correct. The introduction should be expanded.

I have few comments to the manuscript:

1.      Delated extra spaces in all manuscript.

2.      Introduction must be improve.

3.      Discussion is missing, no references to literature.

Taking into account all comments the manuscript may be published in Materials after major revision.

Reviewer 3 Report

Comments and Suggestions for Authors

In this article, the formation mechanism of carbon segregation of medium carbon steel with different combined electromagnetic stirring processes was analyzed. The topic discussed in the article is interesting. There are many mathematical formulas at work.

The work requires several improvements:

1. Due to the importance of the journal, the amount of literature analyzed is small. The subject of carbon steel casting is very extensive and has been used for a very long time.

2. I did not find a description of the us symbol in equation 9.

3. Please improve the quality of Figure 5, especially the miniature charts.

4. The summary should include a short description of the research conducted and only the conclusions drawn from this research. This arrangement will allow readers to relate the results to the adopted methodology.

    After making these changes, the article is ready for publication.    

Author Response

"Please see the attachment

Round 2

Reviewer 2 Report

Comments and Suggestions for Authors

Manuscript can be published in presented form.